# Parasitism Features of a Fig Wasp of Genus *Apocrypta* (Pteromalidae: Pteromalinae) Associated with a Host Belonging to *Ficus* Subgenus *Ficus*

**DOI:** 10.3390/insects14050437

**Published:** 2023-05-03

**Authors:** Po-An Chou, Anthony Bain, Bhanumas Chantarasuwan, Hsy-Yu Tzeng

**Affiliations:** 1Department of Forestry, National Chung Hsing University, No. 145 Xingda Rd., Taichung City 40227, Taiwan; hut23939889@gmail.com; 2Department of Biological Science, National Sun Yat-sen University, No. 70 Lien-Hai Rd., Kaohsiung City 80424, Taiwan; anthonybain22@mail.nsysu.edu.tw; 3Thailand Natural History Museum, National Science Museum, Pathum Thani 12120, Thailand; b.chantarasuwan@gmail.com

**Keywords:** *Apocrypta*, *Ficus pedunculosa* var. *mearnsii*, *Blastophaga pedunculosae*, fig wasp, parasitoid

## Abstract

**Simple Summary:**

Non-pollinating fig wasps (NPFWs) in the genus *Apocrypta* mostly interact with fig species belonging to the subgenus *Sycomorus*. However, an instance of the *Apocrypta* species associated with *Ficus pedunculosa* var. *mearnsii* in the subgenus *Ficus* was observed. To better understand its parasitism features, we inferred its life history with the fig growth and estimated the parasitism ability using a quantitative and qualitative approach that compared the ovipositor sheath-length ratio to the fig gall+wall thickness among the species in *Apocrypta*. Our results showed that this wasp exhibited a high parasitism ability, more advantageous than that of other congeneric species. In contrast, its parasitism rate was low, suggesting that other ecological factors like oviposition strategy and the severe habitat could affect its efficiency. These findings may provide a reference for the symbiosis between fig trees and fig wasps.

**Abstract:**

Non-pollinating fig wasps (NPFWs), particularly long-ovipositored Sycoryctina wasps, exhibit a high species specificity and exert complex ecological effects on the obligate mutualism between the plant genus *Ficus* and pollinating fig wasps. *Apocrypta* is a genus of NPFWs that mostly interacts with the *Ficus* species under the subgenus *Sycomorus*, and the symbiosis case between *Apocrypta* and *F. pedunculosa* var. *mearnsii*, a *Ficus* species under subgenus *Ficus*, is unique. As fig’s internal environments and the wasp communities are distinct between the two subgenera, we addressed the following two questions: (1) Are the parasitism features of the *Apocrypta* wasp associated with *F. pedunculosa* var. *mearnsii* different from those of other congeneric species? (2) Is this *Apocrypta* species an efficient wasp that lives in its unique host? Our observation revealed that this wasp is an endoparasitic idiobiont parasitoid, as most congeneric species are, but developed a relatively long ovipositor. Furthermore, the relationships of the parasitism rate versus the pollinator number, the fig wall, and the sex ratio of the pollinator, respectively, showed that it possessed a higher parasitism ability than that of other congeners. However, its parasitism rate was low, and thus it was not an efficient wasp in its habitat. This difference between parasitism ability and parasitism rate might be a consequence of its oviposition strategy and the severe habitat conditions. These findings may also provide insights into the mechanism to maintain the interaction between the fig tree and the fig wasp community.

## 1. Introduction

Biotic interactions in nature can take a wide range of forms and have complex functions that underpin the entire ecosystem [1,2]. Each interaction involves numerous species that have developed considerable adaptations to maintain an advantage in the evolutionary arms race [3]. This phenomenon is particularly apparent in the symbiosis between Hymenoptera parasitoids and their hosts, with both parties having evolved highly specific morphological and physiological traits. In this situation, clear differences in parasitism features such as life history traits and parasitism ability might be expected if a parasitoid taxon moves to another host and then colonizes a new niche [4,5,6]. Therefore, a comprehensive search for such a phenomenon and a quantification of the consequences that a species brings to the new system is worthwhile.

Fig (*Ficus* spp.) trees are distributed in tropical and subtropical regions, and approximately 750 species have been described to date, with an almost equal split between monoecious and functional dioecious species [7]. A fig (or syconium) is an urn-shaped, nearly closed inflorescence containing tens to thousands of small florets with a single entrance, the ostiole. Figs emit unique odors that attract pollinating fig wasps (Chalcidoidea: Agaonidae; hereafter referred to as “pollinators”), which enter the syconia and lay their eggs in many ovaries to form galls, while the remaining ovaries receive the pollen the wasps have carried and develop into seeds [8]. Thus, figs and pollinators form a species–species obligate nursery pollination mutualism [9,10], except for a few cases of multiple pollinator species associated with a single *Ficus* species [11,12,13,14]. Alongside these pollinators, other chalcid wasps known as non-pollinating fig wasps (NPFWs) also interact with the figs, most of which oviposit from outside the syconium. These NPFWs have a wide range of diets: (1) Gallers compete with pollinators for female florets before or during the pollination period and, like pollinators, produce galls on which their larvae will feed [15,16]. (2) Kleptoparasites lay eggs during the middle- and late-development stages of the syconia in the galls of pollinators or gallers. The parasite larvae compete for the gall food, and the host larva will die of starvation [17]. (3) Parasitoids also lay eggs during the middle- and late- development stages of the syconia, and their larvae directly feed on the larvae of other wasp species [16,17,18,19]. (4) Seed predators, which occur very late in the development of the fig, cannot induce gall formation, with their larvae feeding on developing seeds instead [20,21]. The diet of NPFWs is often inferred from their ovipositing period and the resulting changes in the hatching wasps within a fig [17,18,22], and only a few studies have identified morphologically the diet from the position of the parasite larva in figs [19,23].

Like pollinators, NPFWs have long coevolutionary histories with their hosts. Indeed, NPFWs follow the speciation of their host fig trees more closely than pollinators in Africa [24] and have a similar codiversification history to that of pollinators in Asia [25]. Not only focusing on phylogeny, numerous studies have also directly investigated the corresponding morphological traits between the *Ficus* species and NPFWs, the most striking of which are the fig-wall thickness and the NPFW ovipositor length [22,26,27], as NPFWs need an “appropriate” ovipositor length to reach the resources inside the host fig, which has led to the formation of species-specific relationships. However, many studies to date have found that host-switching between *Ficus* species is a non-negligible phenomenon in NPFWs [28,29,30], suggesting that it may be more common in NPFWs than in other parasitoid species.

The genus *Apocrypta* (Pteromalidae: Pteromalinae) is a group of NPFWs including approximately 30 described species [31,32]. Members of this genus are considered monophyletic [33] and parasitize figs in the subgenus *Sycomorus* with two known exceptions: *A. suprasegmenta* on *F. pustulata* in the subgenus *Ficus* in the Philippines [31]; and an undescribed *Apocrypta* species on *F. bizanae* in the subgenus *Urostigma* in Africa [33]. Here, we describe a new and rare exception to the list with an *Apocrypta* species living in *F. pedunculosa* var. *mearnsii* (called “*Apocrypta* pedunculosa” hereafter) in the subgenus *Ficus,* which is in the same subgenus as *F. pustulata* [7]. To document this example further, we have looked at the parasitism features of “*Apocrypta* pedunculosa” as it is a species living with an unusual host *Ficus*. Consequently, this could be regarded as one of the most conservative genera at high taxonomic ranks between NPFW and *Ficus* species. Other clear examples can be found in *Idarnes* wasps with figs of the subgenus *Urostigma* section *Americana,* and in *Sycophaga* wasps with figs of the subgenus *Sycomorus* [15].

The studied host plant, *Ficus pedunculosa* var. *mearnsii*, is a functional dioecious fig that grows on uplifted coral reef shores between Luzon and southern Taiwan [7,34]. Based on its morphological characters, *F. pedunculosa* belongs to the subgenus *Ficus*, section *Ficus*, subsection *Frutescentiae* [7]. However, phylogenetic research has suggested that it is a more basal relative of subsection *Frutescentiae* [35]. The fig wasp community that symbiotically interacts with the studied fig species includes the pollinator *Blastophaga pedunculosae* and a single NPFW species, “*Apocrypta* pedunculosa” (Figure 1A,B), which in Taiwan is only reared from the figs of this *Ficus* species [36]. Both wasp species were confirmed via the comparison of their related taxonomical descriptions [31,37].

Numerous studies have explored the symbiotic relationships between *Apocrypta* species and fig trees [17,18,22,38,39,40], but most have focused on those species that live in *Ficus* species in the subgenus Sycomorus, with studies that mention *Apocrypta* associated with *Ficus* species in different subgenera being limited to simple observations [31,33]. Although no direct hypothesis has been raised previously for the changes in the life history traits and the parasitism ability of an *Apocrypta* wasp utilizing a host fig other than that of subgenus *Sycomorus*, we consider it is worthy to assume since figs of every *Ficus* subgenera display many morphological differences [7,41,42], that a NPFW species arriving on another fig species will encounter problems if it tries to efficiently parasitize the host. For example, when most *Apocrypta* wasps lay eggs, the fig cavities are filled with fluid in the species belonging to subgenus *Sycomorus* [41], but in the subgenus *Ficus,* the cavities cram male flowers, which might influence the sensitivity of the ovipositor (Appendix A). Moreover, wasp communities between subgenus *Sycomorus* and subgenus *Ficus* might lead to a niche differentiation in *Apocrypta* wasps, which have been reported previously to consume *Ceratosolen* pollinating wasps (Agaonidae: Kradibiinae) or *Sycophaga* non-pollinating wasps (Pteromalidae: Sycophaginae; Appendix A) [17,23,40,43]. In the case of *F. pedunculosa* var. *mearnsii*, however, members of the wasp community are *Blastophaga* pollinating wasps (Agaonidae: Agaoninae) and “*Apocrypta* pedunculosa”. Therefore, the main hypothesis of this study is based on the statements above leading us to think that “*Apocrypta* pedunculosa”, with such a different host environment, will develop different parasitism features of life history traits, parasitism ability, and parasitism rate to be an efficient parasitoid. To verify this hypothesis, the following questions were addressed: (1) Are the parasitism features of “*Apocrypta* pedunculosa” different from those of other congeneric species that have been studied? (2) Is “*Apocrypta* pedunculosa” an efficient wasp that lives in the fig of *F. pedunculosa* var. *mearnsii* in the South of Taiwan?

## 2. Materials and Methods

The selected study sites were Frog Rock Trail (21°56′40.2″ N, 120°46′58.9″ E) and Jialeshuei (21°59′39.6″ N, 120°51′50.9″ E) in Kenting National Park in the southernmost part of Taiwan. These sites are located on an uplifted coral reef coastal area, as *F. pedunculosa* var. *mearnsii* is a frontier species among the coastal vegetation that inhabits areas approximately 2–10 m above sea level [44]. Both sites have a tropical climate, with the rainy season between May and October. During the study period, the mean temperature and precipitation were 24.8 °C and 2198.4 mm, respectively, at Frog Rock Trail and 24 °C and 2414.5 mm, respectively, at Jialeshuei (data obtained from the Central Weather Bureau of Taiwan).

To understand the parasitism features of “*Apocrypta* pedunculosa”, first we needed to consider the ecological features of its host fig and observe its life history traits. As the life history of a fig wasp corresponds to the growth period of its host fig, the clear definition of the fig growth is necessary. Referring to the definition that is adopted most widely [45], we divided the fig growth into five phases:-A phase denotes the initial period before the female florets are mature.-B phase is the short period when the female florets are mature, prompting the pollinator to enter the syconium to lay eggs and pollinate.-C phase is the growth period of the seeds or galls.-D phase denotes the period when the male florets are mature (only on syconia from male trees), at which time the wasp larvae hatch and leave the syconia.-E phase (only on female trees) is the period when the seeds become mature and attract frugivores.

A fig growth survey was conducted from October 2016 to April 2017 on eight randomly selected male fig trees. Six to eight male figs that simultaneously produced male figs in A phase were labeled on each tree and surveyed every two weeks. During each survey, one labeled fig from each tree was collected, and its diameter was measured. To estimate the duration of the development of “*Apocrypta* pedunculosa” larvae, any fig observed with a wasp ovipositing in it during the survey was labeled (Figure 1C,D), and the time required for the “*Apocrypta* pedunculosa” larvae to mature was then recorded.

Among parasitism features adopted to determine the performance of a parasite [46], the factor parasitism ability is one of the most non-negligible variables. To understand the parasitism ability of “*Apocrypta* pedunculosa” within the unique association and determine whether it is different from other congeneric species, both quantitative and qualitative tests were performed in this study. With a quantitative test, the parasitism ability of a NPFW could be reflected by the impact it brought to the figpollinator mutualism. Previous studies had listed different aspects of such impact, including decreasing the abundance of the pollinator [15,34], decreasing the abundance of seed [47,48,49], prompting the fig to develop a thicker wall in the evolutionary arms race [27,42], increasing the sex ratio of the pollinator (proportion of males to the total number of pollinators within a fig) [50,51,52], and causing fig abortion if the plant is over-parasitized [16,49]. Except for seed decrease and fig abortion aspects that are observed only in the mutualism between a monoecious fig and the pollinator, the above aspects of impact are thought to exist, more or less, in our studied mutualism. Furthermore, the parasitism rate (proportion of *Apocrypta* wasps to the total number of fig wasps within a fig) is an indicator commonly used to determine the parasitism ability [53,54]. Thus, here we investigated the relationships between the parasitism rate and variables of those aspects of impact on the *Ficus*–pollinator obligate mutualism, namely the abundance of pollinator, the fig-wall thickness, and the sex ratio of the pollinator to evaluate the parasitism ability of “*Apocrypta* pedunculosa”. For the quantitative analysis, we collected a total of 205 unopened D-phase figs (to ensure that no wasps had left) between March 2015 and December 2016 (137 figs from 32 trees at Frog Rock Trail and 68 figs from 9 trees at Jialeshuei). The number of fig wasps in each fig was then counted and the wall thickness of 117 figs was also measured. A generalized linear mixed model (GLMM) was used to explore the relationships between the parasitism rate and three variables (pollinator abundance, fig-wall thickness, and pollinator sex ratio), with location, monsoon period, and tree ID of the sampled figs included as random factors.

In the qualitative test to determine whether the parasitism ability of “*Apocrypta* pedunculosa” is different from that of other congeneric species, we compared it with four other associations, including species belonging to the subgenus *Sycomorus* (Appendix A). As two of the *Apocrypta* wasps in those associations were directly observed using C-phase figs [55] and almost all *Apocrypta* species in previous studies were described as parasitoid [17,18,22,23,39,40,43], we assumed that the four *Apocrypta* species were laying eggs in mid- C-phase figs of their host where the wall thickness was thought not to be significantly different from that of D-phase figs [27]. Twelve samples of D-phase figs from each species were dissected and the male flowers were removed so that the thicknesses of the fig wall and the wall + gall layer could be measured with a precision of 0.01 mm (Appendix A). Next, five female wasps from each of the *Apocrypta* species associated with these *Ficus* hosts were randomly selected and the lengths of their ovipositor sheaths were measured with a precision of 0.01 mm. The significance of the differences between variables in the comparison was assessed by using the Mann–Whitney U test for two variables and the Kruskal–Wallis test and Dunn’s post hoc test for more than two variables.

In the analysis to decide whether “*Apocrypta* pedunculosa” is an efficient wasp living in the fig of *F. pedunculosa* var. *mearnsii* in the South of Taiwan, the parasitism rate is used again as it is not only an indicator to estimate the impacts of an NPFW on the mutualism but also is widely used to assess the effectiveness of a bio-control agent colonizing the pest population [46,56,57]. Here we counted the parasitism rate of each fig and then ranked it in number distribution to see how efficiently “*Apocrypta* pedunculosa” utilized the figs. Furthermore, to evaluate the exact relationship between the parasitoid and the pollinator, it is necessary to categorize which type of functional response “*Apocrypta* pedunculosa” exhibits in its symbiotic system because functional response not only describes the relationship between the predator (or parasitoid) number and prey density but also indicates the ability of a natural enemy to limit a pest to a low threshold as biocontrol [53,54,58,59]. Since three general types of functional response were known as their distinct relationships (linear, curve with a saturating limit. and sigmoid with a saturating limit) between predator and prey [53,54,60], we not only investigated the functional response of “*Apocrypta* pedunculosa” abundance to the host density (total wasp number per fig) with logistic regression but also visualized its relationship.

All analyses were conducted in R v. 4.0.2. The package lme4 was adopted for the GLMM, assuming Poisson and binomial distributions of the residuals, and a likelihood ratio test was used to evaluate the significance of the fixed variables. Package frair was used for the functional response test [60].

## 3. Results

It took approximately 13 weeks for a fig to develop from the A phase to the D phase, and the largest fig diameter was observed in the 10th week of development (Figure 2). The pollinator *B. pedunculosae* entered the figs to lay eggs during the second and third weeks of male fig development (Figure 2), while the female “*Apocrypta* pedunculosa” oviposited in pollinated C-phase figs during the 7th or 8th weeks. Following oviposition by *Apocrypta* wasps, the figs grew to the D phase in approximately 6 weeks (*n* = 16), indicating that the larvae need 6 weeks to develop.

In the quantitative analysis of parasitism ability, each of the 205 male figs contained an average of 138.50 ± 104.59 (mean ± SD) pollinators, which had a sex ratio of 0.25 ± 0.19. In contrast, the average number of “*Apocrypta* pedunculosa” in each fig was 30.45 ± 39.08 with a sex ratio of 0.49 ± 0.25. The abundance of pollinators was significantly higher than that of “*Apocrypta* pedunculosa” (Mann–Whitney U test, W = 5125, *p* < 0.001), with the latter exhibiting a parasitism rate of 0.21 ± 0.24. The generalized linear model (GLMM) results showed a significant negative correlation between the parasitism rate of “*Apocrypta* pedunculosa” and the abundance of pollinators (Table 1; Figure 3A), which indicated that the *Apocrypta* wasp significantly decreased the pollinators. Table 1 also showed no significant relationship between the parasitism rate and the fig-wall thickness of *F. pedunculosa* var. *mearnsii*, and no strong trend could be observed in Figure 3B. This result meant fig- wall thickness was not an efficient mechanism to impede the *Apocrypta* wasp’s exploitation. Moreover, we could not observe any significant relationship either but a high standard error between the parasitism rate of “*Apocrypta* pedunculosa” and the sex ratio of the pollinator in Table 1 and Figure 3C showed a nearly neutral trend between the two variables. The neutral trend and high standard error revealed a random situation rather than discriminatively distorting the sex ratio of the pollinator when “*Apocrypta* pedunculosa” utilized the galls inside a fig.

The analysis comparing *F. pedunculosa* var. *mearnsii* with four other species in the subgenus *Sycomorus* also revealed that the former species displayed the smallest fig diameter (Kruskal–Wallis H = 26.206, *p* < 0.001), the thinnest wall (H = 51.615, *p* < 0.001), and the thinnest wall + gall layer (H = 48.654, *p* < 0.001; Table 2). In contrast, the length of the “*Apocrypta* pedunculosa” ovipositor sheath was not significantly different from that of other congeneric species, except for *A. bakeri* and *A. caudata*, which are associated with *F. hispida* and *F. variegata*, respectively (Kruskal-Wallis H = 21.305, *p* < 0.001; Table 2).

The distribution of the parasitism rate showed that 84 (41%) of the 205 collected male figs displayed a parasitism rate below 0.05 (Figure 4). In addition, 32 (16%) figs were not utilized by this NPFW, and 88 (43%) reared <10 NPFWs. Also, “*Apocrypta* pedunculosa” exhibited a type II functional response so that its abundance approached saturation with the host density increasing (logistic regression, z = −27.481, *p* < 0.001; Figure 5).

## 4. Discussion

The *Apocrypta* wasp associated with *F. pedunculosa* var. *mearnsii* lay their eggs only in pollinated male figs, and this occurs several weeks after the pollination event, which is similar to other NPFW species in Asia [23] and Africa [17]. Previous research on the oviposition timing and diets of *Apocrypta* wasps [23] showed that two species associated with *F. racemosa* exhibited different diet strategies: *Apocrypta westwoodi* mainly oviposited on late C-phase figs, being an endoparasitic idiobiont, and an undescribed species oviposited on B- to late C-phase figs, displaying a rare feeding habit known as phytoentomophagy. Based on this correlation between the oviposition timing and feeding habits, we considered that *Apocrypta* wasps living in *F. pedunculosa* var. *mearnsii* figs are endoparasitic idiobiont parasitoids, as are most of the species in the genus.

A high parasitism ability in this studied wasp was supported by the quantitative analysis which showed that “*Apocrypta* pedunculosa” could easily overcome the fig-wall barrier in most situations and bring a cost to the male function in the fig–pollinator system. Moreover, we could observe that “*Apocrypta* pedunculosa” did not increase the sex ratio of the pollinator, suggesting its high ability to reach oviposition sites, which allows it to utilize not only the outer galls where more of the pollinator’s female offspring are found but also the inner galls in which more male offspring occur [50,51,52]. Likewise, the qualitative result indicated again that “*Apocrypta* pedunculosa” obtained the advantage of a relatively long ovipositor compared with the relatively thin fig wall and wall + gall layer, which could lead to a higher parasitism ability in its symbiotic interaction than among other congeneric species.

Although a long ovipositor seems to help its owner more easily access the resources from another closely related host [19], an NPFW trying to utilize a new fig host must still conquer a series of difference, including the fig scent, the fig-wall structure, and the internal space of figs [42,61]. In our study, *Ficus pedunculosa* var. *mearnsii* developed a thin fig wall that other *Apocrypta* spp. could overcome. However, none of those *Apocrypta* wasps was seen to colonize it, suggesting that the fig wall might not be the factor restricting *Apocrypta* wasps from shifting. Instead, we thought the fig cavity was more likely to be the factor as “*Apocrypta* pedunculosa” had evolved a stouter body and stronger legs that were considered to help the male shuttle through crowded male flowers for mating while the slender bodies of other *Apocrypta* spp. seemed to hardly move inside (Appendix A). The high parasitism ability and the high prevalence among male figs displayed by “*Apocrypta* pedunculosa” also suggests that this wasp have well adapted the environment and the food of its current fig host. However, our results indicated that despite its strong parasitism ability, “*Apocrypta* pedunculosa” exhibited a low parasitism rate but a high prevalence in the southern Taiwan population of *F. pedunculosa* var. *mearnsii*. Since this wasp is the only NPFW species that parasitizes the figs of this host species, its parasitism rate was easy to calculate but difficult to compare with other *Apocrypta* case studies, in which the parasitism rates were calculated by summing all co-existing NPFW species on a single *Ficus* species [17,22,40]. To our knowledge, the only exception to this is the study which reported a higher parasitism rate in a specific NPFW species (*Sycoscapter* sp.) under glasshouse conditions [62] than what we recorded for “*Apocrypta* pedunculosa” in the present study (0.29 vs. 0.21, respectively), suggesting that this *Apocrypta* species does not exhibit a high parasitism rate despite its increased ability.

We confirmed that “*Apocrypta* pedunculosa” displayed a type II functional response and inverse density-dependence and also suggested that this wasp has a parasitoid feeding habit and utilizes the pollinator directly. Both positive and negative interactions have been found between other *Apocrypta* wasps and their corresponding *Ficus*–pollinator systems in previous studies [17,18,40]. We hypothesized that these different interaction types might result from the complex feeding habits that occur in this NPFW genus [23], which could thus reflect different functional response types in different *Apocrypta* wasps. Since the type Ⅱ functional response and inverse density-dependence are common in parasitoid–host systems [53], many studies have described the factors that lead to this ecological phenomenon, including a limited egg number and long handling time [59,63]. Therefore, we applied these factors to the present NPFW case and established two hypotheses:

(1)Oviposition strategy: *Apocrypta* wasps have been shown to be synovigenic, meaning that they produce and mature eggs continuously over time [64]. This suggests that, no matter how strong the parasitism ability of “*Apocrypta* pedunculosa” is, it cannot overwhelm a single fig with its offspring at any one time. This concept was also directly supported by the two studies which confirmed that *Sycoscapter* wasps, closely related to *Apocrypta* wasps, usually lay only a few eggs in a single fig [65,66]. These physiological and behavioral adaptions are believed to guarantee that parasitoid NPFWs would never overexploit pollinators, as only male pollinators can liberate all the wasps inside by making holes in the enclosed inflorescence [8]. If NPFWs were too abundant in a fig, they would probably kill off all the males, resulting in all the wasps dying inside their natal fig and thus being counter-selected.(2)Severe habitat: Although *Apocrypta* wasps are confirmed to live long, surviving for up to 60 days when feeding in the laboratory [22,64], we inferred that “*Apocrypta* pedunculosa” would not live for such a long period in the field due to the severe climate conditions, such as high temperature and irradiance, strong seasonal monsoon winds, and sporadic typhoons [44,67]. These climate conditions detrimentally affect not only the fig wasps but also the fig trees [67,68,69]. Therefore, even if the *F. pedunculosa* var. *mearnsii* population could supply sufficient resources through continuous fig production and the formation of dense aggregations which might enable “*Apocrypta* pedunculosa” to display high searching efficacy, this wasp would still periodically encounter hazardous weather conditions, which would reduce its parasitism rate among male figs [44,70]. This implies that the studied wasp might have a long handling time with decreased efficiency in this hazardous habitat. Ants are also known to disturb the NPFW oviposition, and an ant species was observed in *F. pedubculosa* var. *mearnsii* figs [71]. However, we believe that this ant would not exert sufficient pressure to cause this parasitism phenomenon, so hazardous climate conditions may be the main influencing factor in this study.

This is the first study to investigate an *Apocrypta* species utilizing a fig species belonging to the subgenus *Ficus*. Our results supported that this NPFW is an endoparasitic idiobiont parasitoid exhibiting high parasitism ability, more advantageous than that of other congeneric species. These findings indicated that a parasitoid fig wasp in a highly conservative genus may change its parasitism features to take more advantage of its current host, compared with other conservative associations, after moving and colonizing a unique new host. In contrast, the parasitism rate of this studied wasp was low despite its high parasitism ability. This incongruence between parasitism ability and parasitism rate might be a consequence of its oviposition strategy and the severe habitat conditions. Since certain fig species have reportedly become keystone species [72], we hope that this study can provide insights into the mechanism that maintains the interactions between fig trees and fig wasp communities or even maintains the bio-system. Moreover, as it has been reported that some fig species have become invasive plants [73], we also hope this study can help address the incongruence between potential parasitism ability and real parasitism rate and provide insights into parasitic ecology and biocontrol application.

## Figures and Tables

**Figure 1 insects-14-00437-f001:**
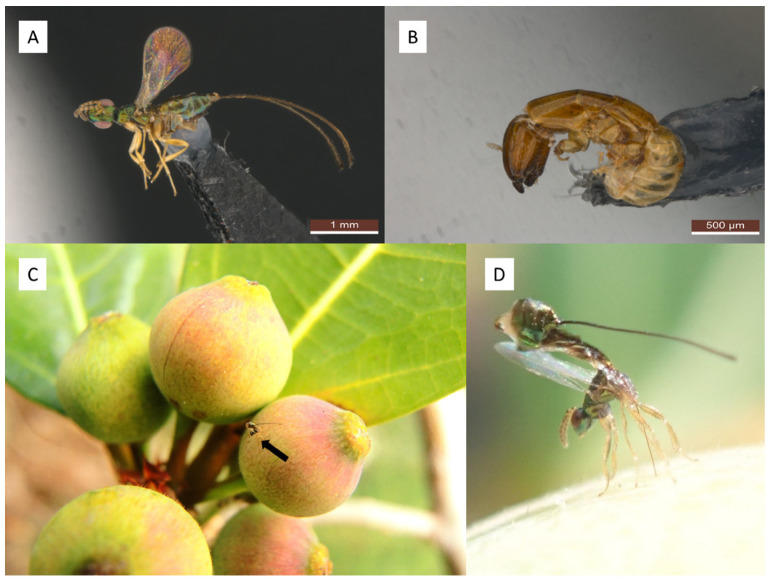
Illustrations of “*Apocrypta* pedunculosa” in *Ficus pedunculosa* var. *mearnsii*. (**A**) Lateral view of female. (**B**) Lateral view of male. (**C**) C phase fig of *F. pedunculosa* var. *mearnsii* with a female “*Apocrypta* pedunculosa” ovipositing (arrow pointed). (**D**) Oviposition posture.

**Figure 2 insects-14-00437-f002:**
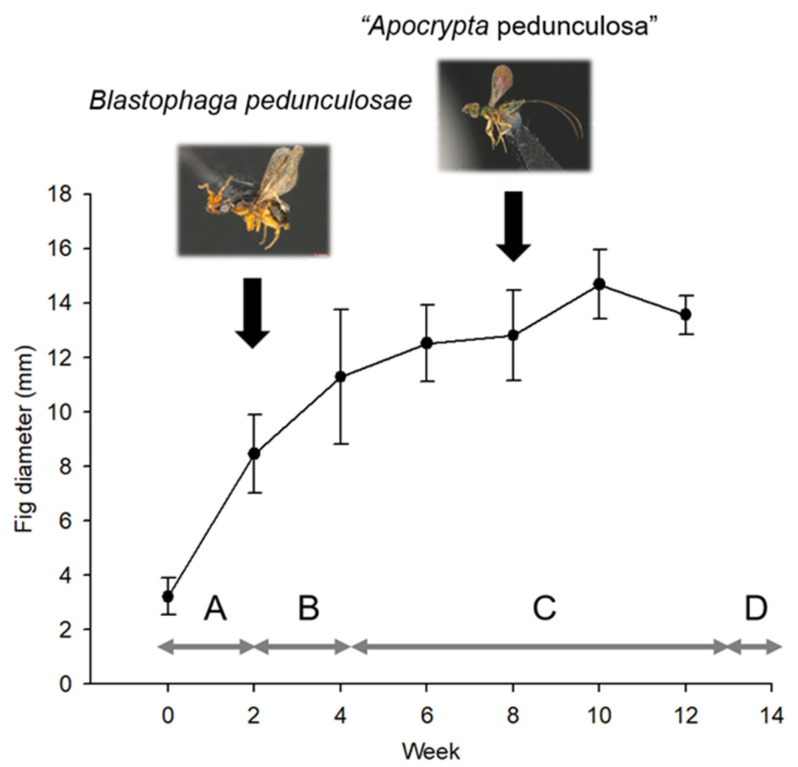
Fig diameter during the development of *Ficus pedunculosa* var. *mearnsii* and the oviposition time of the two fig wasp species. Dots and bars represent the mean ± SD of fig diameters, and letters represent the fig development phases.

**Figure 3 insects-14-00437-f003:**
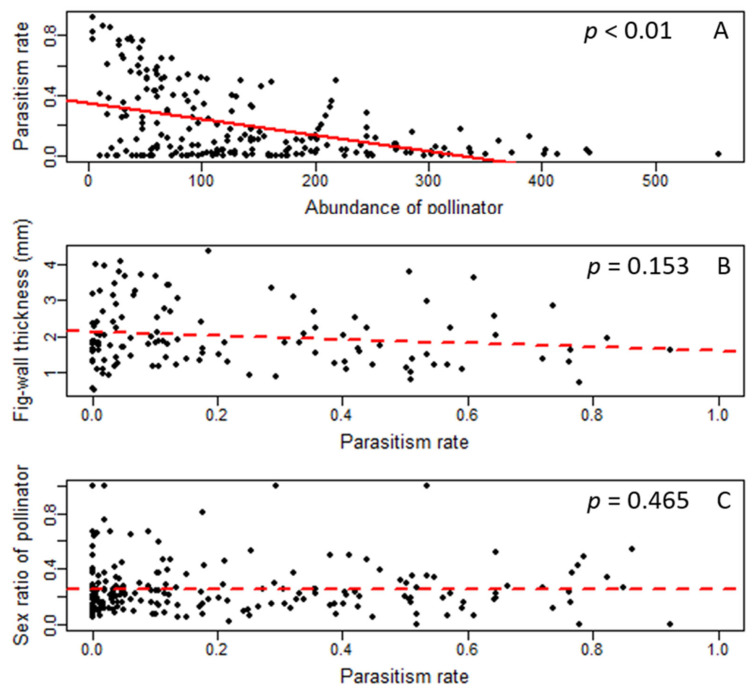
Scatterplot of the relationship between the parasitism rate of “*Apocrypta* pedunculosa” and its effects on the *Ficus*-pollinator system. (**A**): Abundance of pollinator vs. parasitism rate; (**B**): parasitism rate vs. fig-wall thickness; (**C**): parasitism rate vs. sex ratio of pollinator. The solid line and the dashed line indicate, respectively, the significant and non-significant relationships (also shown as *p*-values) in the generalized linear mixed-model results.

**Figure 4 insects-14-00437-f004:**
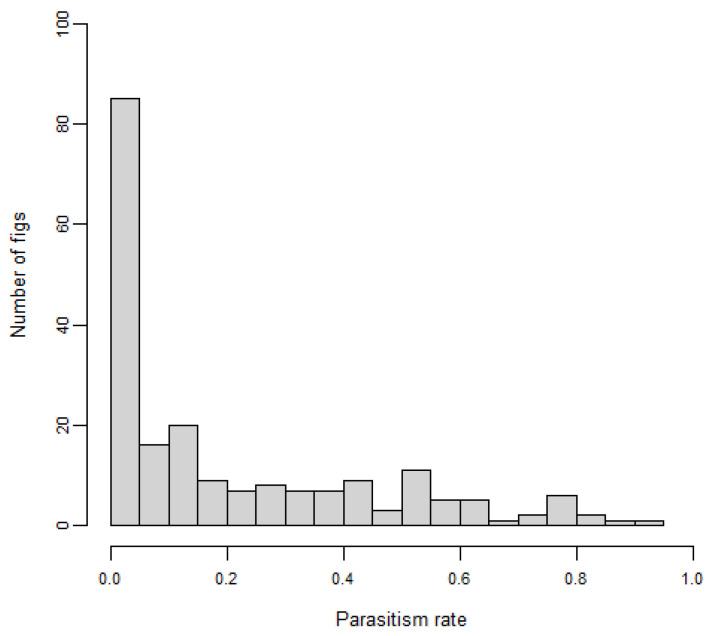
The number of figs at different parasitism rate exhibited by “*Apocrypta* pedunculosa”.

**Figure 5 insects-14-00437-f005:**
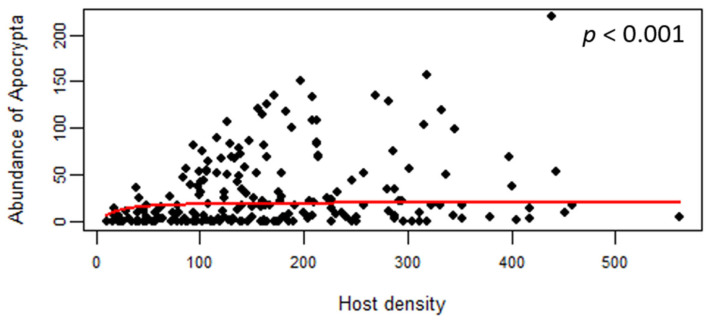
Type Ⅱ functional response trend between “*Apocrypta* pedunculosa” and the host density (total wasp number per fig) of *Ficus pedunculosa* var. *mearnsii*. The solid line and *p*-value indicates the significant relationship in the logistic regression result.

**Table 1 insects-14-00437-t001:** Generalized linear mixed-model results of “*Apocrypta* pedunculosa” on *Ficus pedunculosa* var. *mearnsii* and the pollinator.

Respond Variable	Fixed Effect	N	Estimate	Standard Error	LR Test	
					LR	*p* Value
Abundance of pollinator	Parasitism rate	205	−0.003	0.00002	279.66	<0.01 *
Parasitism rate	Fig wall	117	−0.420	0.310	2.038	0.153
Parasitism rate	Sex ratio of pollinator	205	0.729	0.979	0.533	0.465

Asterisk represents a significant difference.

**Table 2 insects-14-00437-t002:** Comparison results of host fig traits and ovipositor sheath length between “*Apocrypta* pedunculosa” of *Ficus pedunculosa* var. *mearnsii* and other congeneric species.

*Ficus* spp.		Subgen. *Ficus*	Subgen. *Sycomorus*			
		*F. pedunculsoa* var. *mearnsii*	*F. fistulosa*	*F. hispida*	*F. semicordata*	*F. variegata*
Diameter (mm)	Mean ± SD	14.37 ± 2.74	20.63 ± 1.97	32.56 ± 2.12 *	25.63 ± 2.54 *	31.33 ± 1.15 *
	Dunn’s test	-	6.500	22.000	11.500	20.000
	*p*-value	-	0.201	<0.001	0.024	0.001
Wall (mm)	Mean ± SD	1.18 ± 0.12	3.25 ± 0.56 *	4.85 ± 0.74 *	2.54 ± 0.40	3.91 ± 0.45 *
	Dunn’s test	-	24.375	46.000	13.625	36.000
	*p*-value	-	0.001	<0.001	0.056	<0.001
Wall+galls (mm)	Mean ± SD	3.54 ± 0.21	5.99 ± 0.29 *	8.06 ± 0.56 *	6.25 ± 0.84 *	6.69 ± 0.38 *
	Dunn’s test	-	17.667	−47.792	−23.208	−31.333
	*p*-value	-	0.013	<0.001	0.001	<0.001
Ovipositor sheath length of *Apocrypta* spp. (mm)	Mean ± SD	2.26 ± 0.35	3.13 ± 0.23	4.02 ± 0.09 *	2.12 ± 0.05	3.90 ± 0.19 *
	Dunn’s test	-	6.400	15.200	−2.200	12.600
	*p*-value	-	0.169	0.001	0.636	0.007
Ovipositor sheath/Wall+galls		0.64	0.52	0.50	0.34	0.58

Asterisks represent a significant difference from “*Apocrypta* pedunculosa” or from *Ficus pedunculosa* var*. mearnsii*.

## Data Availability

The data that support the findings of this study are available from the corresponding author, H.-Y.T., upon reasonable request.

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
