# Peer review of "Parasitism Features of a Fig Wasp of Genus *Apocrypta* (Pteromalidae: Pteromalinae) Associated with a Host Belonging to *Ficus* Subgenus *Ficus"

_insects, 2023, doi:10.3390/insects14050437_

Round 1
Reviewer 1 Report
The topic is interesting and deserves attention. Reading the manuscript is clear and easily understood, aims are clearly defined and the experimental design is appropriate.
In my opinion, Materials and methods should be described more schematically and less discursive while the aims of the study described at the end of the introduction. For this. I suggest minor revision
Line 51: Remove the term “symbiotic”.
Line 127-135: I suggest moving this part in the “Introduction”
Line 176,177: It is not clear how “ parasitism rate” is defined, you should add a reference.
Line 196-201: Unclear, you should add more information. How to “categorize” the functional response?
Line 202-204: These statements generally do not appear in M&M. I suggest moving this part at the end of the manuscript, for example under “Declarations”.
Line 222: Please correct B. pedunculosae
Table 1: please correct “parasitism rate”
Line 240: The value of sex ratio refers to male or female?
Line 239-242: The Standard deviation is very high. when there is so much variability, it is useful to look at the standard error.
Author Response
Response to the comments from Reviewer #1
The topic is interesting and deserves attention. Reading the manuscript is clear and easily understood, aims are clearly defined and the experimental design is appropriate.
Reply: Thank you for your positive review of our manuscript.
In my opinion, Materials and methods should be described more schematically and less discursive while the aims of the study described at the end of the introduction. For this. I suggest minor revision
Reply: Thank you for you kind remarks. We have added more description to make the Material and methods part clearer. The paragraphs in Meterial and methods have been also re-arranged to make the link with Introduction stronger. L151-155, L169-229.
Minor points:
Line 51: Remove the term “symbiotic”.
Reply: Thanks. We have removed the term. L51.
Line 127-135: I suggest moving this part in the “Introduction”
Reply: Thanks. We have moved this part in Introduction. L100-108.
Line 176,177: It is not clear how “ parasitism rate” is defined, you should add a reference
Reply: Thanks. We have added desceiption about “ parasitism rate” with references. L182.
Line 196-201: Unclear, you should add more information. How to “categorize” the functional response?
Reply: Thanks. We have added more description about functional response and how we categorize it. L221-225.
Line 202-204: These statements generally do not appear in M&M. I suggest moving this part at the end of the manuscript, for example under “Declarations”.
Reply: Thanks. These statements are moved to the part “Declarations” . L420-422.
Line 222: Please correct B. pedunculosae
Reply: Thanks. The term has been corrected. L223.
Table 1: please correct “parasitism rate”
Reply: Thanks. The term has been corrected. Table 1.
Line 240: The value of sex ratio refers to male or female?
Reply: Thanks. We have added desceiption about “ sex ratio” with references. L178.
Line 239-242: The Standard deviation is very high. when there is so much variability, it is useful to look at the standard error.
Reply: Thanks. We have added more description about this result with standard error included. L248-249, L251-258.
Reviewer 2 Report
Dear authors,
After a careful review of your work, I have identified several areas that require minor revisions to enhance the clarity and quality of your manuscript.
Please provide more detailed information about the statistical methods used in your study. Specifically, please include the sample size, type of statistical analysis performed, and any relevant statistical software used.
The literature review section could benefit from additional references to recent studies in the field to provide a more comprehensive overview of the topic.
Please clarify the implications and significance of your findings more explicitly in the discussion section. Provide specific examples of how your results can inform future research.
I appreciate the effort and time that you have put into this manuscript, and I believe that addressing these minor revisions will greatly improve the overall quality of your work.
Sincerely,
Author Response
Response to the comments from Reviewer #2
After a careful review of your work, I have identified several areas that require minor revisions to enhance the clarity and quality of your manuscript.
Reply: Thank you for your positive review of our manuscript.
Please provide more detailed information about the statistical methods used in your study. Specifically, please include the sample size, type of statistical analysis performed, and any relevant statistical software used.
Reply: Thank you for you kind remarks. We have added more description to make the Material and methods part clearer. The paragraphs in Meterial and methods have been also re-arranged to make the link with Introduction stronger. The sample size, statistical analyses and software were added in L187-194, L202-209, L221-229.
The literature review section could benefit from additional references to recent studies in the field to provide a more comprehensive overview of the topic
Reply: Thank you for your kind remarks. We have added more references and more description to make this manuscript clearer.
Please clarify the implications and significance of your findings more explicitly in the discussion section. Provide specific examples of how your results can inform future research.
Reply: Thank you for your kind remarks. We have added more description about our findings in L398-403 and L406-409.
I appreciate the effort and time that you have put into this manuscript, and I believe that addressing these minor revisions will greatly improve the overall quality of your work.
Reply: Thnak you again for your positive review of our manuscript.